# Multiplex detection of "*Candidatus* Liberibacter asiaticus" and *Spiroplasma citri* by qPCR and droplet digital PCR

**Yogita Maheshwari**[1☯¤a], **Vijayanandraj Selvaraj**[1☯¤b]*, **Kristine Godfrey**[2],
**Subhas Hajeri**[3], **Raymond Yokomi**[1]*

**1** San Joaquin Valley Agricultural Sciences Center, Agricultural Research Service, United States Department of Agriculture, Parlier, California, United States of America, **2** Contained Research Facility, University of California, Davis, Davis, California, United States of America, **3** Citrus Pest Detection Program, Central California Tristeza Eradication Agency, Tulare, California, United States of America

☯ These authors contributed equally to this work.
¤a Current address: Plant Virus Laboratory, Council of Scientific and Industrial Research-Institute of Himalayan Bioresource Technology, Palampur, Himachal Pradesh, India
¤b Current address: Plant Molecular Virology Laboratory, Council of Scientific and Industrial Research—National Botanical Research Institute, Lucknow, Uttar Pradesh, India
* ray.yokomi@usda.gov (RY); vijayanandraj@hotmail.com (VS)

**Data Availability Statement:** All relevant data are within the manuscript and its Supporting Information files.

## Abstract

"*Candidatus* Liberibacter asiaticus" (*C*Las) and *Spiroplasma citri* are phloem-limited bacteria that infect citrus and are transmitted by insect vectors. *S. citri* causes citrus stubborn disease (CSD) and is vectored by the beet leafhopper in California. *C*Las is associated with the devastating citrus disease, Huanglongbing (HLB), and is vectored by the Asian citrus psyllid. *C*Las is a regulatory pathogen spreading in citrus on residential properties in southern California and is an imminent threat to spread to commercial citrus plantings. CSD is endemic in California and has symptoms in citrus that can be easily confused with HLB. Therefore, the objective of this study was to develop a multiplex qPCR and duplex droplet digital PCR (ddPCR) assay for simultaneous detection of *C*Las and *S. citri* to be used where both pathogens can co-exist. The multiplex qPCR assay was designed to detect multicopy genes of *C*Las—RNR (5 copies) and *S. citri*–SPV1 ORF1 (13 copies), respectively, and citrus cytochrome oxidase (COX) as internal positive control. Absolute quantitation of these pathogens was achieved by duplex ddPCR as a supplement for marginal qPCR results. Duplex ddPCR allowed higher sensitivity than qPCR for detection of *C*Las and *S. citri*. ddPCR showed higher tolerance to inhibitors and yielded highly reproducible results. The multiplex qPCR assay has the benefit of testing both pathogens at reduced cost and can serve to augment the official regulatory protocol for *C*Las detection in California. Moreover, the ddPCR provided unambiguous absolute detection of *C*Las and *S. citri* at very low concentrations without any standards for pathogen titer.

## Introduction

Citrus is severely affected by fastidious vascular colonizing bacteria such as "*Candidatus* Liberibacter asiaticus" (*C*Las) and *Spiroplasma citri* (*S. citri*). The fastidious bacteria are introduced

**Funding:** RY 5300-191 Citrus Research Board RY Yok-18 California citrus Nursery Board RY 2034-22000-013-10D United States Department of Agriculture, Agricultural Research Service, Base Project. The funders had no role in study design, data collection and analysis, decision to publish, or preparation of the manuscript.

**Competing interests:** The authors have declared that no competing interests exist.

directly into phloem sieve tubes by phloem feeding insect vectors. *C*Las is a Gram-negative, α–proteobacterium [1] associated with the devastating citrus disease Huanglongbing (HLB), known as citrus greening. *C*Las is transmitted by the Asian citrus psyllid (ACP), *Diaphorina citri* Kuwayama (Hemiptera: Psyllidae). HLB is widely being distributed in Asia, South Africa, Central America, South America and some parts of United States viz., Florida, Texas and California [2]. *S. citri* is a wall-less Gram-positive bacteria that causes citrus stubborn disease (CSD) [3]. CSD is an endemic disease and widely distributed in semi-arid regions of California, where citrus is grown mostly as an irrigated crop [4]. *S. citri* is transmitted by the beet leafhopper, *Neoaliturus* (*Circulifer*) *tenellus* (Baker) (Hemiptera: Cicadellidae) [5] in the United States and *Circulifer haematoceps* (Mulsant & Rey) (Hemiptera: Cicadellidae) [6] in the Mediterranean region.

HLB and CSD have latent periods of several months to a year or more. Symptoms of HLB and CSD can be easily confused with each other and nutritional disorders [7] (Fig 1). In general, both the diseases are difficult to diagnose and differentiate at the early stages of infection. Fruit symptoms include irregular shape or small lopsided fruits with varying size and maturity on the same tree. During later stages of infection, the plant shows twig decline, stunted growth, low yield and in case of HLB, eventually leads to death of tree (Fig 1).

Detection of these diseases are challenging due to seasonal fluctuation and sporadic distribution of bacterial titer within the tree [8–10]. The spread of HLB into commercial citrus trees and presumptive co-infection with *S. citri* is eminent with the widespread of distribution of ACP and establishment of HLB in residential properties of southern California. Although CSD reduces tree vigor and contributes to loss of production, dual infection may cause a more

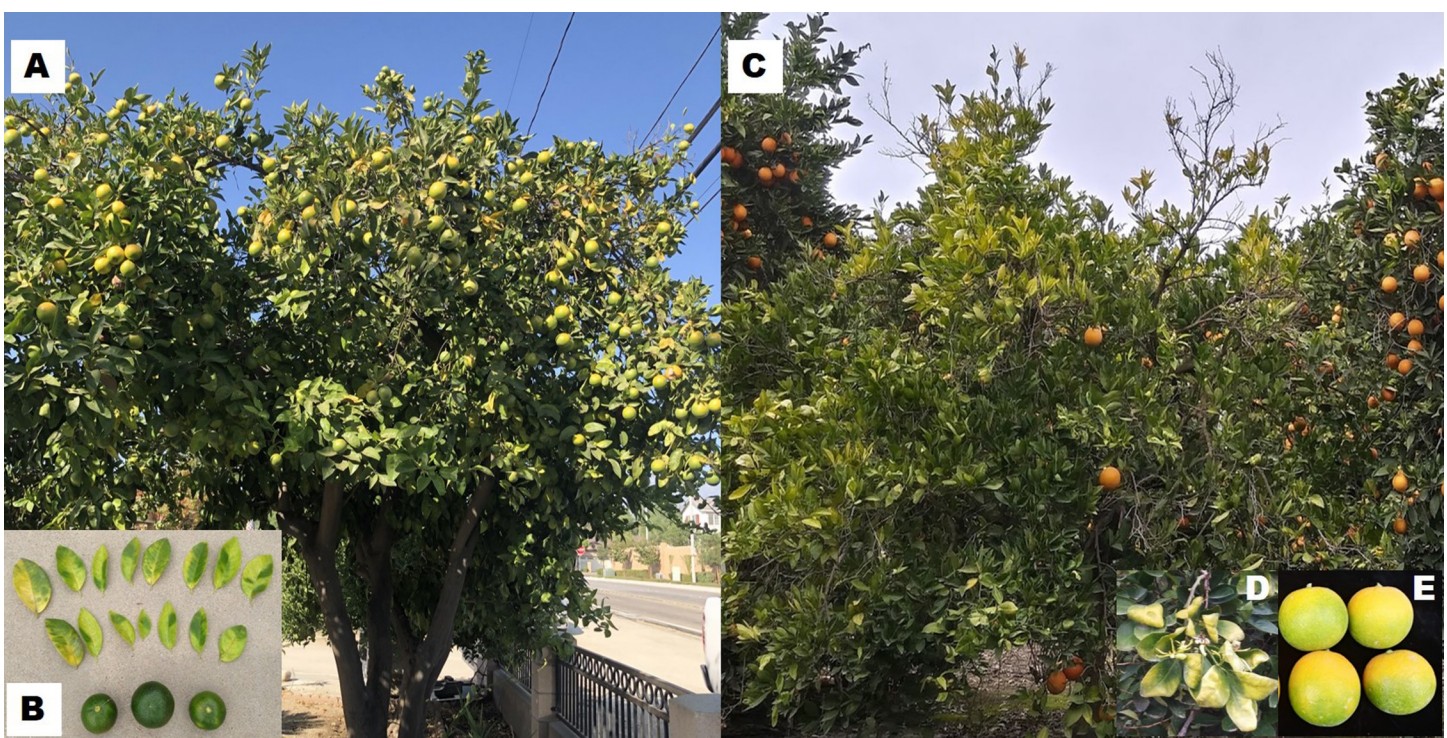

**Fig 1.** Comparison of qPCR-confirmed Washington navel tree infected with Huanglongbing (HLB) (Left) and Citrus stubborn disease (CSD) (Right). (A) *C*Las infected tree in suburban southern California; (B) Blotchy mottle leaves and distorted fruits symptoms of HLB. (C) *S. citri* infected tree (left) next to a healthy tree (right) in a field in central California; (D) Chlorotic leaves with shortened internodes and (E) smaller misshapen fruit, with seasonal stylar-end greening typical of CSD. HLB pictures provided by Magally Luque-Williams, CDFA.

rapid and deadly tree demise. In a greenhouse study, coinfection of these bacterial pathogens led to severe yellowing, dieback symptoms and later the plant died within 18 months post inoculation (dpi) in sweet orange, while the plants infected only with *C*Las or *S. ctiri* survived during the period observed [11].

A grower-funded citrus pest detection program (CPDP) surveys the commercial citrus in central California. HLB survey is of high priority to CPDP. Field inspectors visually inspect every tree on the perimeter of a grove for HLB symptoms and the ACP. Since HLB symptoms are similar to CSD symptoms and CSD is endemic in survey areas of CPDP, accurate diagnosis of HLB and CSD is very critical for implementation of timely control measures. Misdiagnosis of these diseases can lead to false corrective measures and unnecessary regulatory actions. Therefore, a multiplex detection of two bacterial pathogens is needed to distinguish these pathogens in a timely and cost-effective manner. Nucleic acid-based techniques such as real time quantitative PCR (qPCR) and droplet digital PCR (ddPCR) are currently available tools for early detection of *C*Las and *S. citri* in singleplex reactions. Detection of *C*Las utilizes 16S and RNR genes either by qPCR or ddPCR [12–15], whereas, *S. citri* was detected using spiralin SP1 and prophage ORF1 genes by qPCR or ddPCR [4, 10, 16, 17]. In this study, a multiplex qPCR assay was developed and validated for simultaneous detection of *C*Las and *S. citri* and included the citrus COX gene as an internal control. In addition, a duplex ddPCR was developed for absolute quantification of *C*Las and *S. citri* at very low copy numbers without the use of a standard curve.

## Material and methods

### Pathogens and DNA isolation

Citrus tissues infected with *C*Las and *S. citri* were obtained from the Contained Research Facility, University of California, Davis; ARS-USDA, Parlier and citrus fields in the San Joaquin Valley. Total DNA was extracted from citrus tissues by the Cetrimonium bromide (CTAB) method [18]. Nucleic acid quality and quantity was measured using Qubit 3.0 (Thermo Fisher Scientific, USA).

### Primers and probes

Primers and probes used in qPCR and ddPCR are listed in Table 1. TaqMan probes were synthesized by labeling the 5' terminal nucleotide with 6-carboxy-fluorescein (FAM), VIC and

**Table 1. Primer and probe sequences used for detection of "*Candidatus* Liberibacter asiaticus" and *Spiroplasma citri*.**

| Organism | Target Gene | Primer/Probe name | Sequence (5'-3') | Amplicon length | Reference |
|---|---|---|---|---|---|
| "*Candidatus* Liberibacter asiaticus" | nrdB, β-subunit of ribonucleotide reductase (RNR) | RNR F | CATGCTCCATGAAGCTACCC | 80 bp | [13] |
| | | RNR R | GGAGCATTTAACCCCACGAA | | |
| | | RNR P | 6FAM/CCTCGAAATCGCCTATGCAC/ MGB/NFQ | | |
| *Spiroplasma citri* | SPV1 ORF1 Prophage (ORF 1) | ORF1 F | TGGCAGTTTTGTTTAGTCATCC | 190 bp | [10] |
| | | ORF1 R | GGGTCTAAACGCCGTTAAAGT | | |
| | | ORF1 P | VIC/TTGGGTTTGGTTATTCCATT/ MGB/NFQ | | |
| **Citrus** | Cytochrome oxidase subunit 1 (COX)[a] | COX F | GTATGCCACGTCGCATTCCAGA | 68 bp | [12] |
| | | COX R | GCCAAAACTGCTAAGGGCATTC | | |
| | | COX-P | TexasRed/ATCCAGATGCTTACGCTG G/ MGB/NFQ | | |

[a]Internal control gene for multiplex qPCR assay.

Texas Red for RNR, ORF1 and COX genes, respectively, and the 3' terminal nucleotide with Minor groove binder/nonfluorescent quencher (Thermo Fisher Scientific, USA).

## Multiplex quantitative PCR

The multiplex qPCR-based detection of *C*Las and *S. citri* was carried out in duplex and triplex assay. Duplex qPCR assays are currently used for routine detection of *C*Las and *S. citri* with RNR/COX and ORF1/COX primers and probes, respectively. The triplex qPCR assay was developed in this study to detect RNR/ORF1/COX genes using tenfold serial dilutions of dually infected citrus DNA. The efficiency of triplex qPCR assay was later compared with duplex assay. The primer and probe concentration in triplex reaction mixture for RNR F/R was 0.15 μM and 0.08 μM, respectively; ORF1 F/R and probe was 0.30 μM and 0.15 μM, respectively; COX F/R was 0.30 μM and 0.15 μM, respectively. The reaction mixture contained 10 μl of 2x Perfecta qPCR ToughMix with low ROX and 1 μl of infected citrus DNA template at ~400 ng/μl. The qPCR was performed in CFX96 Real-Time System (Bio-Rad, USA). The thermal cycling consisted of initial denaturation of 3 min at 95°C, followed by 40 cycles of 95°C for 15 sec and 63°C for 40 sec. Two sets of duplex qPCR reactions were performed for detection of RNR/COX and ORF1/COX using tenfold serial dilutions of *C*Las and *S. citri* infected DNA, respectively.

## Plasmid construction

The RNR (80 bp) gene was amplified from DNA extracted from HLB-infected citrus leaves using RNR F/R primers. The prophage gene ORF1 (533 bp) was amplified from *S. citri* DNA Prophage ORF1 F/R primers. The amplicons were ligated in pGEM-T Easy vector (Promega) and transformed in JM-109 (Promega). The positive plasmids were linearized using SpeI restriction enzyme (New England Biolabs, UK) and the concentrations were measured using dsDNA high sensitivity assay kit in Qubit 3.0 fluorometer (Thermo Fisher Scientific, USA). Ten-fold serial dilutions of the linearized plasmids were made to assess analytical sensitivity, linearity, and dynamic range of ddPCR in singleplex and duplex ddPCR assays.

## Thermal gradient optimization of duplex ddPCR

The optimal annealing temperature for duplex ddPCR assay with RNR and ORF1 primers was determined by a thermal gradient in the S1000™ Thermal cycler (Bio-Rad, USA). The temperature range were 53°C, 53.7°C, 55.1°C, 57°C, 59.2°C, 61.1°C, 62.4°C and 63°C. The reaction mixture contained the same amount of *C*Las and *S. citri* DNA with primers/probes concentrations of 0.9 μM/ 0.25 μM. Optimization of duplex ddPCR assay with RNR and ORF1 primers was achieved using linearized plasmids. The duplex ddPCR reaction mixture (20 μl) contained 2x ddPCR Supermix for probes (no dUTP) (BioRad, USA), 0.9 μM of ORF1 and RNR primers, 0.25 μM of ORF1 and RNR probes, and 1 μl of plasmid DNA. The singleplex ddPCR reaction contained the same amount of primers and probes as in duplex assay for *C*Las or *S. citri*. The 20 μl reaction mixture and 70 μl of droplet generation oil was used for generation of emulsion droplets in QX 200 droplet generator (BioRad, USA) using DG8 cartridge. The droplet emulsions were loaded in a 96-well PCR plate and sealed with pierceable foil using a PX1 PCR plate sealer (Bio-Rad, USA). PCR amplification was carried out in a C1000™ thermal cycler. The thermal cycling conditions consisted of 10 min initial denaturation at 95°C, followed by 40 cycles of denaturation at 94°C for 30 sec and annealing/extension at 57°C for 1 min, with a ramp of 2°C/sec. and a final 10 min incubation at 98°C for enzyme deactivation. After thermal cycling, the plate was placed in a QX 200 droplet reader (Bio-Rad, USA) for analyzing each individual droplet by a detector.

### Assessing inter-assay and intra-assay variability of duplex ddPCR

The inter-assay and intra-assay variation of duplex ddPCR assay was carried out using *C*Las and *S. citri*, dual infected leaf DNA samples. The inter-assay variation was determined by measuring the copy number of *C*Las and *S. citri* between three different assays in triplicate. The intra-assay variation was determined by measuring the copy number of *C*Las and *S. citri* within the assay in three replications. Coefficient of Variation (CV) was calculated by standard deviation/mean.

### Estimation of tolerance to residual matrices on duplex ddPCR

The tolerance of duplex ddPCR assay for inhibitors in citrus leaf samples was evaluated with reactions containing different concentration of inhibitors in citrus. The reaction was spiked with same amount of *S. citri* and *C*Las plasmids DNA (ORF1 and RNR) and different quantities of citrus healthy leaf extract (1 to 5μl). The effect of inhibitors in leaf extract were assessed relative to the mean measured signals in each sample with no added inhibitors. Healthy citrus leaf extracts were used to estimate the tolerance of duplex ddPCR assays.

### Data analysis

The qPCR data were analyzed by the BioRad CFX Manager 3.1 software. The ddPCR data were analyzed with QuantaSoft analysis software version 1.7 (Bio-Rad, USA). The positive droplets with amplified products were discriminated from negative droplets by applying a threshold above the negative droplets. Reactions with more than 10,000 accepted droplets per well were used for analysis. The linear regression and P-value of the ddPCR assay were determined by plotting the measured copies of ddPCR and comparing them with expected values of serial dilution of plasmid DNA, *C*Las/*S. citri* infected citrus DNA in Excel. The Poisson error and total error was obtained using QuantaSoft software.

## Results

### Real time quantitative PCR

The triplex qPCR assay regression curve showed good linearity with ten-fold dilutions of *C*Las and *S. citri* infected leaf DNA for RNR ($R^2$ = 0.9983), ORF1 ($R^2$ = 0.9988) and COX ($R^2$ = 0.9992). The PCR efficiency was 100.16, 101.11 and 100 for RNR, ORF1 and COX, respectively (Fig 2). Duplex assay for detection of *C*Las showed a linearity of 99.4% and 99.5%, with an efficiency of 101.6% and 97.14% for RNR and COX, respectively (Fig 3A). *S. citri* duplex detection assay showed linearity of 106.85% and 105.9% for ORF1 and COX, respectively (Fig 3B). Comparison of triplex and duplex assay Ct values for RNR, ORF1 and COX values did not show significant changes for detection of *C*Las and *S. citri*. Detection limit of both triplex and duplex assays was 0.04 ng of total DNA from dually infected with *C*Las and *S. citri*.

### Optimization of duplex ddPCR

The optimum annealing temperature for *C*Las and *S. citri* was selected based on the fluorescence amplitude that differs from the positive and negative droplets in duplex ddPCR assay. An annealing temperature of 57°C was chosen for RNR and ORF1 gene primers and probes for the subsequent duplex ddPCR experiments (Fig 4).

The linear regression curve was obtained by plotting the $Log_{10}$ transformed values between the serially diluted plasmid DNA and expected values in singleplex and duplex ddPCR assays. The RNR and ORF1 plasmid DNA showed $R^2$ = 1 in singleplex assay and $R^2$ = 1 and $R^2$ = 0.999 respectively, in duplex assay. The sensitivity of RNR and ORF1 was 3 copies and 2 copies

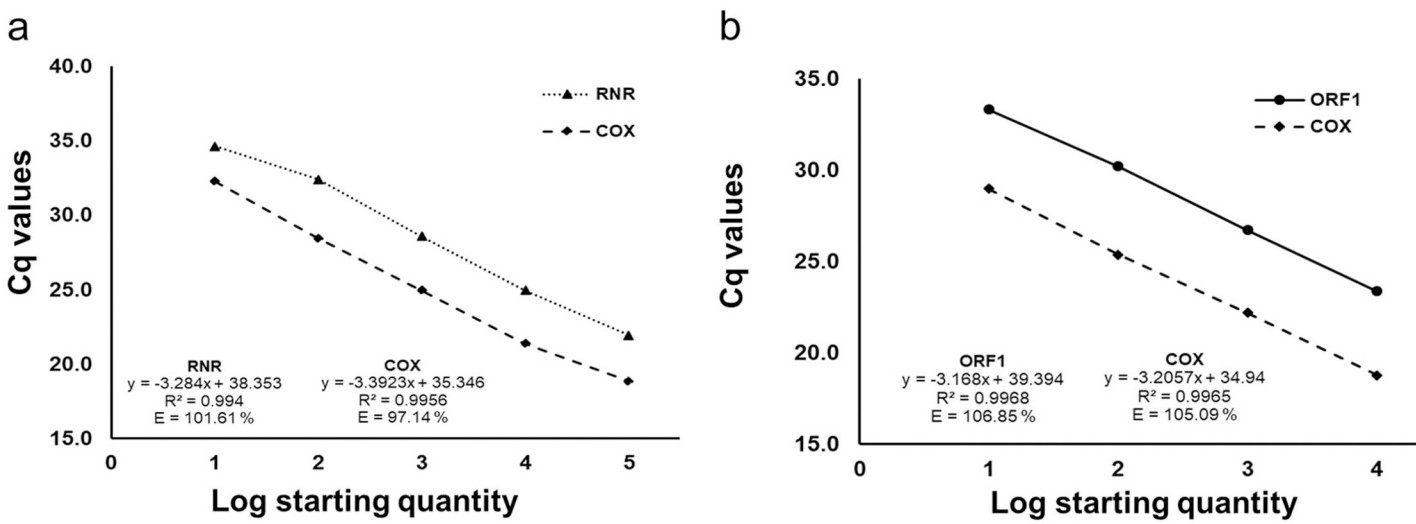

**Fig 2. Calibration curve of triplex qPCR assays with tenfold serial dilution of "*Candidatus* Liberibacter asiaticus" and *Spiroplasma citri* infected DNA (400 ng to 0.004 ng).** Gene specific targets were RNR (dotted line) and ORF1 (unbroken line) for detection of *C*Las and *S. citri*, respectively, and COX (dash line) for citrus DNA.

**Fig 3. Calibration curve of duplex qPCR assays with tenfold serial dilution of "*Candidatus* Liberibacter asiaticus" and *Spiroplasma citri* DNA (400 ng to 0.004 ng) performed using RNR (dotted line), ORF1 (unbroken line) and COX (dash line) gene specific primers.** (a) Detection of *C*Las by duplex qPCR assay using RNR and COX gene; (b) Detection of *S. citri* by duplex qPCR assay using ORF1, and COX gene for citrus DNA.

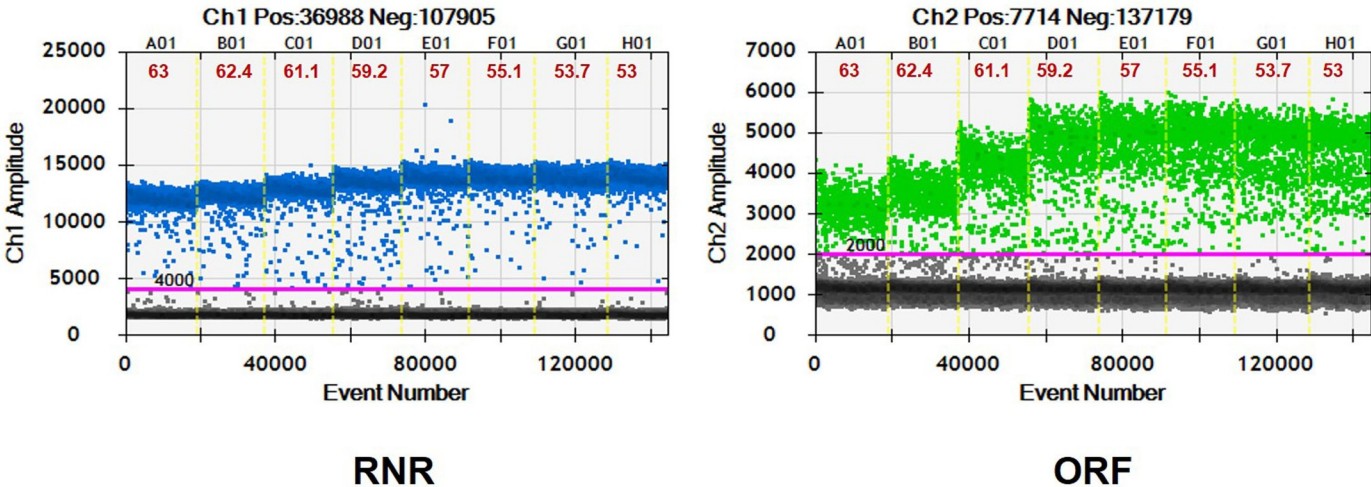

**Fig 4. Thermal gradient droplet digital PCR for optimizing annealing temperature.** (A) RNR of "*Candidatus* Liberibacter asiaticus"; (B) ORF1 of *Spiroplasma citri*. Eight ddPCR reactions with an annealing temperature gradient ranging from 53°C to 63°C are divided by vertical dotted yellow lines. The pink line is the threshold, above which are positive droplets (blue/Green) and below are negative droplets (gray) without any target DNA.

in singleplex assay. The sensitivity of RNR and ORF1 was 2 copies in duplex ddPCR assay (Fig 5; S1 Table).

Singleplex and duplex ddPCR assay of *C*Las and *S. citri* dual infected DNA showed five and four orders of magnitude respectively, between the target input amounts and ddPCR measured values. The RNR and ORF1 primers showed good linearity of $R^2 = 1$ and $R^2 = 0.999$ respectively, in singleplex assay and $R^2 = 1$ in duplex assay. The sensitivity of *C*Las in dual infected citrus leaf DNA was 5 copies and 11 copies in singleplex and duplex ddPCR assay, respectively. The sensitivity of *S. citri* in dual infected citrus leaf DNA was 19 copies and 15 copies in singleplex and duplex ddPCR assay, respectively (S2 Table). There was an excellent correlation (P<0.0001) in copy number of *C*Las and *S. citri* between singleplex and duplex ddPCR assays with linearized plasmids and dual infected citrus leaf DNA (Fig 6).

### Repeatability and reproducibility of duplex ddPCR assay

The absolute quantification of *C*Las and *S. citri* in ten-fold dilution of dual infected citrus leaf DNA produced good repeatability (intra-assay) and reproducibility (inter-assay). The co-efficient of variation of RNR in inter-assay was better compared to intra-assay especially in low titer samples. The co-efficient of variation of ORF1 was better in inter and intra assay (Fig 7).

### Influence of residual matrices on duplex ddPCR assay

The tolerance of duplex ddPCR assay to citrus leaf extract was estimated using RNR and ORF1 primers (Fig 8). The fluorescent signals of positive droplets and negative droplets gradually decreased and increased respectively, with the increasing amount of citrus extract for both *C*Las and *S. citri*. In contrast, the other parameter affected by the presence of the residual matrices for the ddPCR were *S. citri* and *C*Las titer, which decreased with the increasing amount of citrus extract.

### Discussion

*C*Las and *S. citri* are pathogens that colonize the phloem tissues and induce similar symptoms in citrus. Both pathogens are unevenly distributed and the seasonal fluctuation in titer makes the early detection of pathogens challenging. Misdiagnosis of *C*Las would lead to costly

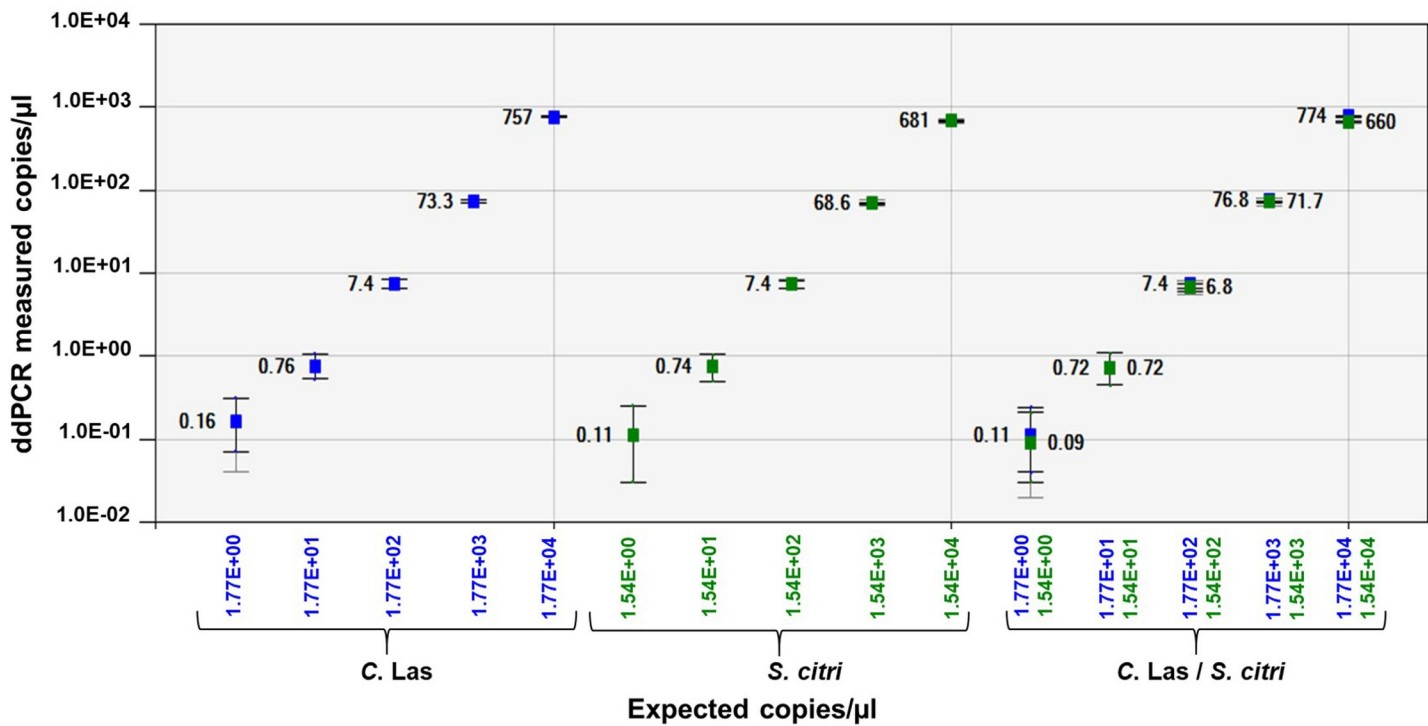

**Fig 5. Linear regression of the singleplex and duplex ddPCR assays for RNR plasmid DNA and ORF1 plasmid DNA for detection of "*Candidatus* Liberibacter asiaticus" (*C*Las) and *Spiroplasma citri* (*S. citri*) respectively.** The Pearson correlation coefficient of singleplex RNR plasmid DNA regression curve (y = 0.8521x-11.405) is 1 and ORF1 plasmid DNA (0.7658x + 6.2768) is 1, respectively. Pearson correlation coefficient of duplex RNR plasmid DNA regression curve (0.871x - 4.3816) is 1 and ORF1 plasmid DNA (0.7414x + 28.637) is 0.999, respectively. The inner error bars indicate the Poisson 95% confidence interval (CI) and the outer error bars show the total 95% CI of replicates. (P<0.0001).

regulatory actions and impact the grower. RNA and DNA targets multiplexing by qPCR analysis has been shown for citrus pathogens [19–24]. Duplex ddPCR has been reported for simultaneous detection of two gene targets for unambiguous detection of *C*Las [14] and *S. citri* [10]. However, concurrent detection of *C*Las and *S. citri* by qPCR and ddPCR is lacking. In this study, sensitive multiplex qPCR technique was developed for simultaneous detection of these two pathogens using RNR (*C*Las), ORF1 (*S. citri*) and COX (citrus internal control) gene primers. A duplex ddPCR was also developed for absolute quantification and detection of these bacterium in dually infected samples.

In areas where both HLB and CSD occur, the multiplex assay developed in this study tests for both pathogens simultaneously without sacrificing efficacy and loss of sensitivity. Inconclusive results from qPCR Ct values at the upper limits is always a critical concern. In such cases, duplex ddPCR assay can provide unambiguous detection and absolute quantification of *C*Las, *S. citri* or both without need of standards [10, 14]. ddPCR was more sensitive compared to qPCR with citrus tissue dually infected with *C*Las and *S. citri*. qPCR detected *C*Las and *S. citri* up to 0.004 ng of total DNA, whereas ddPCR detected up to a dilution of 0.002 ng for *S. citri* and 0.0002 ng for *C*Las.

While ddPCR is robust, reliable, and tolerant to PCR inhibitors, the cost for ddPCR is ~2.3x greater than qPCR due to expendables like gaskets, cartridges and droplet oil needed to test samples each 96-well plate. Additionally, the time required for a ddPCR run is ~2.3x longer than qPCR to assay the same samples in a 96-well plate. Notably, qPCR is a real time assay; whereas ddPCR is an end point assay. Therefore, the best application for ddPCR is as a secondary test to confirm results of questionable qPCR results.

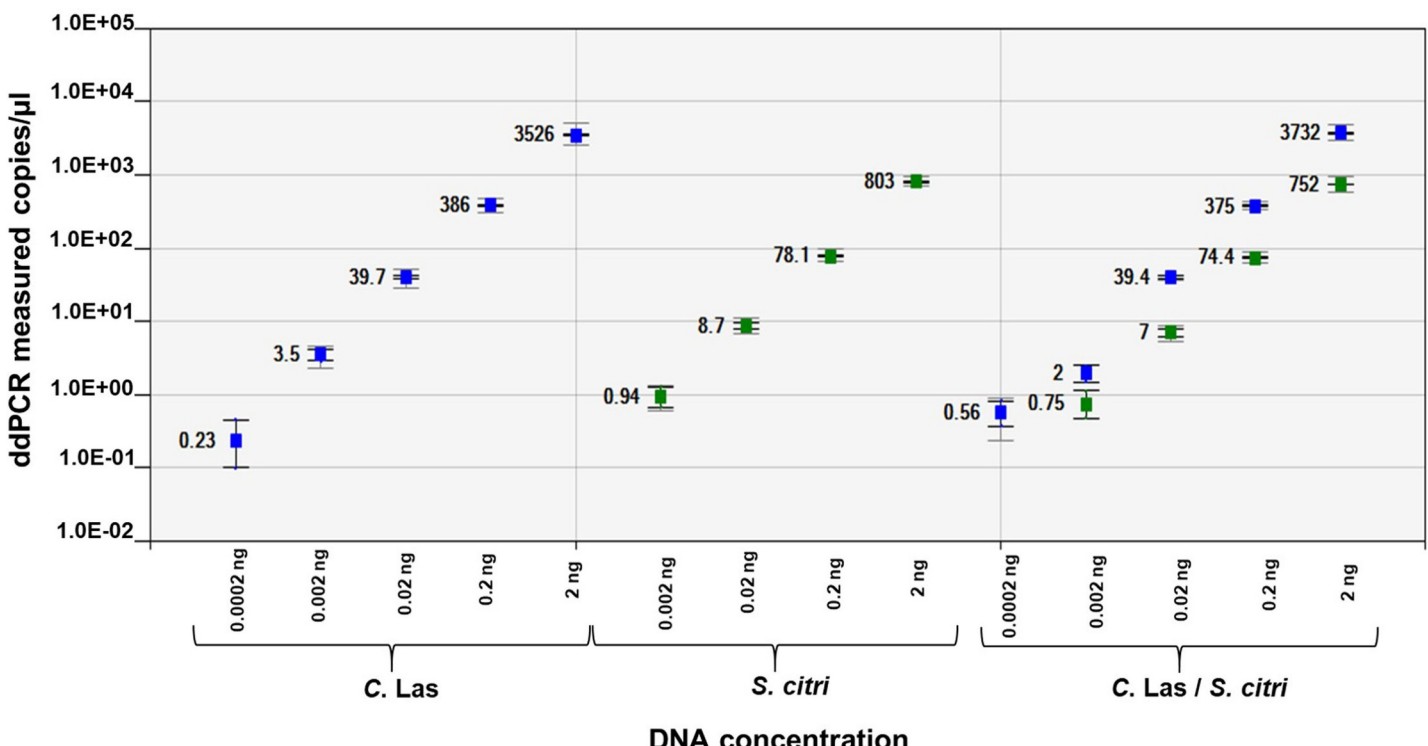

**Fig 6. Linear regression of the singleplex and duplex ddPCR assays for detection of "*Candidatus* Liberibacter asiaticus" (RNR gene) and *Spiroplasma citri* (ORF1 gene) in dually infected citrus leaf DNA.** The Pearson correlation coefficient of singleplex RNR and ORF 1 DNA regression curve is 0.999 and 1, respectively. Pearson correlation coefficient of duplex RNR and ORF1 DNA regression curve is 1. The inner error bars indicate the Poisson 95% confidence interval (CI) and the outer error bars show the total 95% CI of replicates. (P<0.0001).

Previous qPCR and ddPCR methods have been essentially for CLas [12, 14] and *S. citri* [10, 17]. This report combines these protocols to test for both pathogens in a duplex reaction from the same sample and well. Multiplex detection of *CLas* and *S. citri* will be useful in California

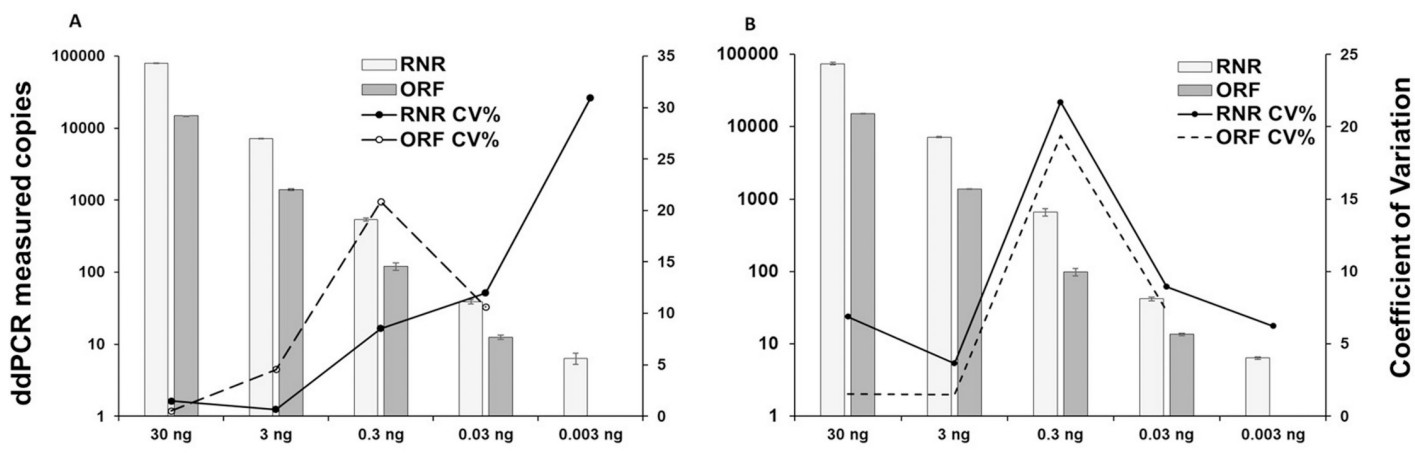

**Fig 7.** Intra-assay (A) and Inter-assay (B) variation of duplex ddPCR assay for detection of RNR gene of "*Candidatus* Liberibacter asiaticus" and ORF1 gene of *Spiroplasma citri* in ten-fold dilutions of dual infected citrus leaf DNA. Bar represents the average of triplicate ddPCR values of each dilutions. CV means coefficient of variation.

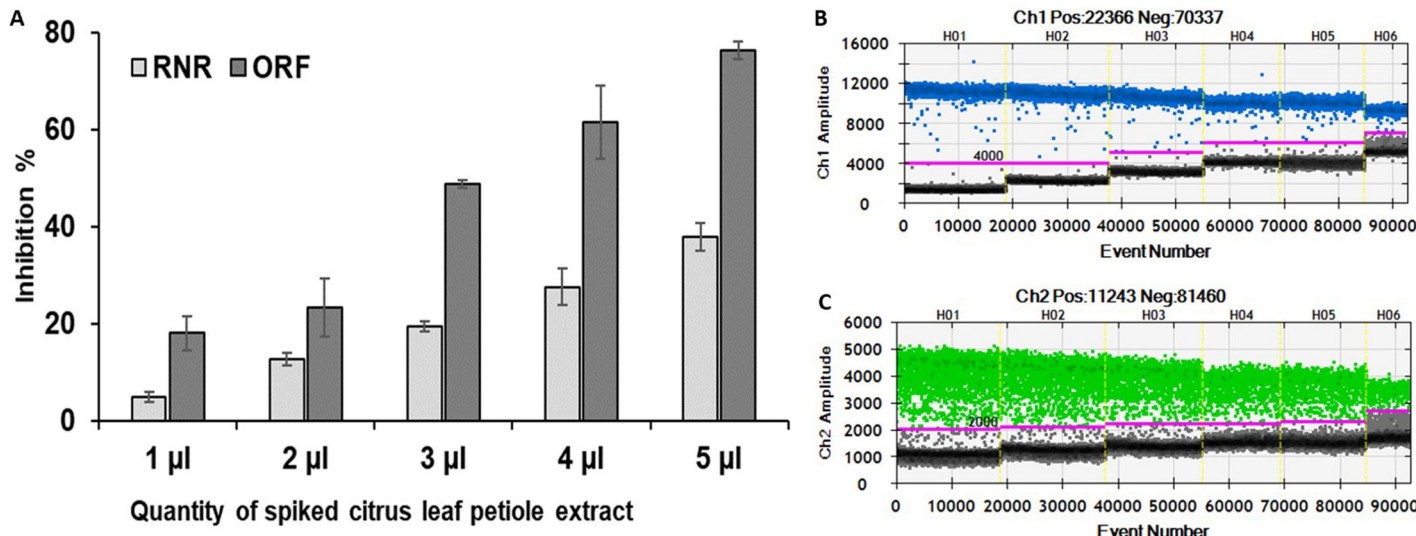

**Fig 8. Influence of citrus leaf petiole extract on quantification of "*Candidatus* Liberibacter asiaticus" and *Spiroplasma citri* by duplex ddPCR assays for RNR gene and ORF1 gene respectively.** (A) ddPCR reaction mixture was spiked with different quantity of citrus leaf petiole extract and equal amount of citrus leaf DNA. Error bars denote standard error of inhibition between three replicates of each reaction. (B, C) one dimensional plot showing only one of three replications for RNR gene and ORF1 gene with citrus leaf petiole extracts.

as HLB spreads from established citrus trees in residential properties to commercial citrus groves. High incidence of CSD has been reported to range from 4 to 60% in certain San Joaquin Valley orchards based on qPCR results [16]. Additional *S. citri* infected trees will likely be found with additional sampling to account for seasonality and erratic pathogen titer. Moreover, *S. citri* is not a regulated pathogen in California and CSD-affected citrus trees are rarely removed. Therefore, presence of CSD must be accounted for during testing of *C*Las in HLB-suspect trees where the outcome is eradication or rouging of infected trees.

## Supporting information

**S1 Table. Quantitative data of "*Candidatus* Liberibacter asiaticus" and *Spiroplasma citri* plasmid DNA with RNR and ORF1 primers in ddPCR assays.**
(DOCX)

**S2 Table. Quantitative data of "*Candidatus* Liberibacter asiaticus" and *Spiroplasma citri* dual infected citrus leaf DNA with RNR and ORF1 primers in ddPCR assays.**
(DOCX)

## Acknowledgments

We thank Robert DeBorde (United States Department of Agriculture-Agricultural Research Service, San Joaquin Valley Agricultural Sciences Center, Parlier, CA) for technical assistance. Mention of trade names or commercial products in this publication is solely for providing specific information and does not imply recommendation or endorsement by the USDA. USDA is an equal opportunity provider and employer.

## Author Contributions

**Conceptualization:** Yogita Maheshwari, Vijayanandraj Selvaraj, Raymond Yokomi.

**Data curation:** Yogita Maheshwari, Vijayanandraj Selvaraj.

**Formal analysis:** Yogita Maheshwari, Vijayanandraj Selvaraj.

**Funding acquisition:** Raymond Yokomi.

**Investigation:** Yogita Maheshwari, Vijayanandraj Selvaraj, Raymond Yokomi.

**Methodology:** Yogita Maheshwari, Vijayanandraj Selvaraj, Subhas Hajeri, Raymond Yokomi.

**Project administration:** Raymond Yokomi.

**Resources:** Kristine Godfrey, Subhas Hajeri, Raymond Yokomi.

**Software:** Raymond Yokomi.

**Supervision:** Raymond Yokomi.

**Writing – original draft:** Yogita Maheshwari, Vijayanandraj Selvaraj, Raymond Yokomi.

**Writing – review & editing:** Kristine Godfrey, Subhas Hajeri.

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
