## [Decision Letter · Decision Letter 0]

17 Feb 2021

PONE-D-20-34072

Multiplex detection of “Candidatus Liberibacter asiaticus” and Spiroplasma citri by qPCR and droplet digital PCR

PLOS ONE

Dear Dr. Yokomi,

Thank you for submitting your manuscript to PLOS ONE. After careful consideration, we feel that it has merit but does not fully meet PLOS ONE’s publication criteria as it currently stands. Therefore, we invite you to submit a revised version of the manuscript that addresses the points raised during the review process.

We look forward to receiving your revised manuscript.

Kind regards,

Zonghua Wang, Ph.D.

Academic Editor

PLOS ONE

Journal Requirements:

Reviewers' comments:

Reviewer's Responses to Questions

**Comments to the Author**

1. Is the manuscript technically sound, and do the data support the conclusions?

Reviewer #1: Yes

Reviewer #2: Yes

2. Has the statistical analysis been performed appropriately and rigorously? 

Reviewer #1: Yes

Reviewer #2: I Don't Know

3. Have the authors made all data underlying the findings in their manuscript fully available?

Reviewer #1: Yes

Reviewer #2: Yes

4. Is the manuscript presented in an intelligible fashion and written in standard English?

Reviewer #1: Yes

Reviewer #2: Yes

5. Review Comments to the Author

Reviewer #1: The manuscript presented by Maheshwari et al presents the multiplex detection of “Candidatus Liberibacter asiaticus ” and Spiroplasma citri by qPCR and droplet digital PCR. This study contains major results that deserve to be published and made available to the scientific community. The writing needs to be better focused on the main objectives instead of being descriptive. In discussion section, one additional paragraph could be added to explain any new parameter or technical improvement by comparison with previous researches, as a number of papers reporting sensitive detection techniques for Clas and S. citri.

Reviewer #2: I think that qPCR and droplet digital are good methods for Candidatus Liberibacter asiaticus and Spiroplasma citri detection. If they were used to detect in the field, do you compare the cost of droplet and qPCR? Which one is more economic? The primer of COXf was lost 3 bases as the reference, does the amplification efficiency is the same?

6. PLOS authors have the option to publish the peer review history of their article (what does this mean?). If published, this will include your full peer review and any attached files.

Reviewer #1: **Yes: **Huasong Zou

Reviewer #2: No

---

## [Author Response · Author response to Decision Letter 0]

25 Feb 2021

Reviewer 1 

• The writing needs to be better focused on the main objectives instead of being descriptive. The research objective is stated more clearly in Abstract Page 2, Lines 29-31. To improve focus, clarity was added in the Discussion Page 13, Lines 270-271; Lines 279-282; Page 14, Lines 287-295; Lines 298-302. The major data presented, however, needs to stay as written as documentation for the efficacy and reliability of the assays which are needed if the procedures are to be used for regulatory samples.

• In discussion section, one additional paragraph could be added to explain any new parameter or technical improvement by comparison with previous researches, as a number of papers reporting sensitive detection techniques for Clas and S. citri. Although there are numerous non- PCR-based assays developed for CLas and S. citri detection, the regulatory standard is still qPCR. ddPCR was added to serve as secondary test for more precise pathogen detection, albeit, more costly and less high throughput. Some text was added in Discussion, Page 14, Lines 293-295; Lines 298-302.

Reviewer 2

• If they were used to detect in the field, do you compare the cost of droplet and qPCR? Which one is more economic? Comparative costs and time to complete ddPCR versus qPCR was added in discussion, Page 14, Lines 287-292.

• The primer of COXf was lost 3 bases as the reference, does the amplification efficiency is the same? Thank you for pointing out this disparity. It was a typo and page 6 Table 1 has been corrected.

---

## [Editor Report · Decision Letter 1]

3 Mar 2021

Multiplex detection of “Candidatus Liberibacter asiaticus” and Spiroplasma citri by qPCR and droplet digital PCR

PONE-D-20-34072R1

Dear Dr. Yokomi,

We’re pleased to inform you that your manuscript has been judged scientifically suitable for publication and will be formally accepted for publication once it meets all outstanding technical requirements.

Kind regards,

Zonghua Wang, Ph.D.

Academic Editor

PLOS ONE
---

## [Editor Report · Acceptance letter]

9 Mar 2021

PONE-D-20-34072R1 

Multiplex detection of “*Candidatus* Liberibacter asiaticus” and *Spiroplasma citri* by qPCR and droplet digital PCR 

Dear Dr. Yokomi:

I'm pleased to inform you that your manuscript has been deemed suitable for publication in PLOS ONE. Congratulations! Your manuscript is now with our production department. 

Kind regards, 

on behalf of

Prof. Zonghua Wang 

Academic Editor

PLOS ONE